# Review of the Impact of Biofilm Formation on Recurrent *Clostridioides difficile* Infection

**DOI:** 10.3390/microorganisms11102525

**Published:** 2023-10-10

**Authors:** Daira Rubio-Mendoza, Adrián Martínez-Meléndez, Héctor Jesús Maldonado-Garza, Carlos Córdova-Fletes, Elvira Garza-González

**Affiliations:** 1Facultad de Medicina, Universidad Autónoma de Nuevo León, Monterrey 64460, Mexico; dairaguadalajararubiomendoza@gmail.com (D.R.-M.); hectormaldonadog@yahoo.com (H.J.M.-G.); carlos.cordovafl@uanl.edu.mx (C.C.-F.); 2Facultad de Ciencias Químicas, Universidad Autónoma de Nuevo León, San Nicolás de los Garza 66455, Mexico; adrian.mtz.fcq@gmail.com

**Keywords:** *Clostridioides difficile*, recurrent infection, antibiotics, minimum inhibitory concentration

## Abstract

*Clostridioides difficile* infection (CDI) may recur in approximately 10–30% of patients, and the risk of recurrence increases with each successive recurrence, reaching up to 65%. *C. difficile* can form biofilm with approximately 20% of the bacterial genome expressed differently between biofilm and planktonic cells. Biofilm plays several roles that may favor recurrence; for example, it may act as a reservoir of spores, protect the vegetative cells from the activity of antibiotics, and favor the formation of persistent cells. Moreover, the expression of several virulence genes, including TcdA and TcdB toxins, has been associated with recurrence. Several systems and structures associated with adhesion and biofilm formation have been studied in *C. difficile*, including cell-wall proteins, quorum sensing (including LuxS and Agr), Cyclic di-GMP, type IV pili, and flagella. Most antibiotics recommended for the treatment of CDI do not have activity on spores and do not eliminate biofilm. Therapeutic failure in R-CDI has been associated with the inadequate concentration of drugs in the intestinal tract and the antibiotic resistance of a biofilm. This makes it challenging to eradicate *C. difficile* in the intestine, complicating antibacterial therapies and allowing non-eliminated spores to remain in the biofilm, increasing the risk of recurrence. In this review, we examine the role of biofilm on recurrence and the challenges of treating CDI when the bacteria form a biofilm.

## 1. Introduction

*Clostridioides difficile* is a Gram-positive sporulated bacterium that was considered to be the most common cause of healthcare-associated infective diarrhea over the last decades. However, it has been shown that reservoirs of *C. difficile* in the community may participate in the transmission of this infection [1].

CDI is transmitted by the consumption of spores and is associated with the disruption of the gut microbiota by the use of antimicrobials [2,3,4].

A *C. difficile* (CDI) infection ranges from simple colitis to pseudomembranous and fulminant colitis. The symptomatology of CDI is characterized by the presence of at least three loose or unformed stools in 24 h or less, along with a history of antibiotic exposure or evidence of megacolon or severe ileus with a positive laboratory diagnostic test result or colonoscopic or histopathological findings revealing pseudomembranous colitis [2]. The most common risk factors for CDI are age above 65, previous use of antibiotics, recent hospitalization [5,6], and enteral feeding [7] (Table 1).

Vancomycin and metronidazole are first-line CDI treatments [8]. Approximately 10–30% of patients develop recurrent CDI (R-CDI) [9], and the risk of recurrence increases with each successive recurrence, from 40% up to 65% [2]. An R-CDI infection is defined by symptoms within eight weeks after a resolved primary infection. The cause of R-CDI may be a relapse, infection by the same strain, reinfection, or infection by a different strain [2]. In patients with R-CDI, other alternatives, such as the fidaxomicin of a fecal microbiota transplant, may be used [8,10].

Some risk factors of R-CDI include immunosuppression [11]; infection with ribotype 027 5, 078, or 244 [11]; a previous history of CDI; severe CDI [4]; gastrointestinal intervention [6]; ≥15 days of acid-suppressive therapy [7]; and serum albumin levels of ˂2.5 g/dL [5,12] (Table 1). The biofilms in the gut (only by *C. difficile* or with other bacterial species of the microbiota) may contribute to recurrence [13]. Within the biofilm, interactions between bacteria, including *C. difficile* adhesion and chemotaxis, modulation of LuxS/AI-2 quorum sensing (QS) system activity, and regulation of intestinal bile acid levels [13], may have an impact on CDI [13].

**Table 1 microorganisms-11-02525-t001:** Risk factors for primary *Clostridioides difficile* infection (CDI) and recurrent *Clostridioides difficile* infection (R-CDI).

Risk Factor	Odds Ratio	*p*	Ref.
***Primary CDI* ^a^**			
Age ≥ 65 years	2.4 (1.6–3.5) ^c^	<0.001	[6]
Recent hospitalization	2.1 (1.5–3.1) ^b^	<0.001	[6]
Enteral feeding	2.9 (2.0–4.1)	<0.001	[7]
Vascular surgery	2.4 (1.3–4.5)	0.003	[6]
Surgery in the preceding 12 weeks	1.7 (1.3–2.4) ^c^	<0.001	[6]
Surgery and gastrointestinal interventions	1.9 (1.2–3.0) ^b^	0.003	[6]
Myocardial infarction	1.7 (1.2–2.6)	0.003	[6]
Gastrointestinal intervention	2.9 (1.9–4.6)	<0.001	[6]
Congestive heart failure	1.9 (1.4–2.5)	<0.001	[6]
Chronic kidney disease	1.9 (1.4–2.6)	<0.001	[7]
Peripheral vascular disease	1.5 (1.1–2.2)	0.011	[6]
Diabetes with organ damage	2.1 (1.3–3.4)	0.001	[6]
Cerebrovascular disease	2.0 (1.2–3.5)	0.008	[6]
Dementia	4.0 (2.4–6.8)	<0.001	[7]
Connective tissue disease	3.3 (1.6–6.7)	<0.001	[7]
Inflammatory bowel disease	2.2 (1.2–4.1)	0.006	[7]
Urinary tract infection	2.2 (1.3–4.1)	0.004	[6]
Dementia	2.5 (1.3–4.8)	0.003	[6]
Chronic obstructive pulmonary disease	2.0 (1.1–3.8)	0.021	[6]
Leukemia	2.3 (1.2–4.1)	0.004	[7]
Charlson Comorbidity Index score 3 ≥ 3 vs. <3	1.5 (1.2–1.9)	<0.001	[7]
Duration of acid-suppressive therapy (≥15 days)	3.8 (2.9–4.8)	<0.001	[7]
Chemotherapy	1.8 (1.1–2.8)	0.006	[7]
Corticosteroids	1.6 (1.2–2.1)	<0.001	[7]
Immunosuppressant agent use	1.6 (1.22.14)	<0.001	[7]
Proton pump inhibitor use	1.7 (1.3–2.4) ^c^	0.001	[6]
At least one antibiotic (any class)	1.3 (1.1–1.4) ^b^	<0.001	[6]
Cephalosporins	2.2 (1.3–3.8) ^b^	0.003	[6]
	2.1 (1.7–2.7)	<0.001	[7]
Third generation	5.4 (3.0–9.8)	<0.001	[7]
Fourth generation	2.0 (1.4–2.9)	<0.001	[7]
Glycopeptides	3.2 (2.2–4.6)	<0.001	[6]
Fluoroquinolones	1.4 (1.1–2.1)	0.022	[6]
	1.6 (1.2–2.1)	<0.001	[7]
Meropenem	1.7 (1.2–2.6)	0.003	[6]
Carbapenem	4.8 (3.6–6.5)	<0.001	[7]
Clindamycin	2.0 (1.4–2.9)	<0.001	[7]
Aminoglycoside	2.7 (1.7–4.1)	<0.001	[7]
Tetracycline	2.9 (1.3–6.3)	0.005	[7]
Linezolid	2.3 (1.3–4.2)	0.003	[7]
Rifampicin	3.4 (1.2–9.4)	0.013	[7]
Total duration of antibiotic therapy (≥15 days)	3.7 (2.9–4.8)	<0.001	[7]
***R-CDI* ^a^**			
Age ≥ 65 years	1.6 (1.1–2.3)	0.0012	[14]
Additional non-CDI antibiotics during follow-up	4.2 (2.1–8.5)	0.001	[14]
Proton pump inhibitors during follow-up	2.14 (1.1–4.0) 1.6 (1.4–1.9)	0.019 NR	[14,15]
Nasogastric tube insertion	8.7 (1.2–59.1)	0.026	[16]
Cardiovascular disease	3.0 (1.2–7.3)	0.015	[17]
Immunosuppressive comorbidities	3.8 (1.3–11.2)	0.012	[17]
Dementia	3.2 (1.2–8.4)	0.014	[17]

^a^ Only risk factors were reported. Odds ratio values are included. ^b^ European multicenter, prospective, biannual, point-prevalence study of *Clostridium difficile* infection in hospitalized patients with diarrhea (EUCLID). ^c^ CDI cases/controls were identified from a single center in Germany (parallel study site).

## 2. Pathophysiology of *C. difficile* Infection

The CDI begins with the oral consumption of spores, which survive the acidic pH of the stomach and pass to the large intestine, where they interact with the primary and secondary bile acids that define whether the spore germinates or continues as a spore [18]. Once the spores germinate, the vegetative cells release enzymes, including collagenase, hyaluronidase, chondroitin sulfatase, enterotoxin A (TcdA), and cytotoxin (TcdB). These enzymes damage the cytoskeleton of intestinal cells [19], causing the condensation of cellular actin [20] and breaking the tight cell junctions between cells. This leads to fluid loss, local inflammation [19,21], and the destruction of the surface epithelium, causing the rounding and detachment of epithelial cells [20]. This process is associated with the infiltration of neutrophils into the submucosa and is mediated by several cytokines, including tumor necrosis factor alpha and interleukin-1β [22].

Additionally, it results in mast cell activation and the production of reactive oxygen species, and a pseudomembrane composed of neutrophils, fibrin, mucin, and cellular debris develops over the intestinal mucosa [21]. Overall, the extent of colonic damage is associated with the toxin concentration and duration of exposure before diagnosis [20]. A summary of the pathophysiology of CDI is presented in Figure 1.

## 3. Virulence Factors: Toxins and Spores

Virulence factors are structures or strategies of microbes that contribute to colonization and survival and may cause damage to the host [23]; they include secretory proteins such as toxins and spores.

Toxigenic strains of *C. difficile* contain a 19.6 kb chromosomal region known as the pathogenicity locus (PaLoc). This region comprises the genes encoding toxin A or TcdA (*tcdA* gene) and toxin B or TcdB (*tcdB* gene), as well as the accessory genes for TcdR, TcdE, TcdL, and TcdC proteins (*tcdR*, *tcdE*, *tcdL*, and *tcdC* genes). Toxin A and toxin B are glucosyltransferases with four domains, including an amino-terminal glucosyltransferase domain, an autoprotease, a translocation/pore-forming domain, and a C-terminal combined repetitive oligopeptide repeat domain (CROPS) [19,24,25]. Toxins A and B secreted in the colon bind to cell-surface glycans through the CROPS domain [25].

Approximately 20% of *C. difficile* isolates obtained from non-outbreak cases produce a third toxin called binary toxin, or *C. difficile* transferase (CDT) [26]. This toxin is encoded by the genes *cdtA* and *cdtB*, which are located in a 6.2 kb region known as the CDT locus (CdtLoc) [27,28]. CdtA is an ADP-ribosyl transferase that acts on actin, and CdtB forms a pore in acidified endosomes and facilitates the transfer of CdtA to the cytosol [27,28]. *C. difficile* is classified into 34 toxinotypes (I–XXXIV) based on changes in the PaLoc and CDT genes [29,30]. The toxinotypes represent changes in toxins A and B and, thus, the differences in functional properties and virulence [21,30,31,32].

Worldwide, the most frequent toxinotype in humans is toxinotype III (RT027- *tcdA*+ *tcdB*+ *cdtA*, and *cdtB*+), followed by toxinotype IV (RT023- *tcdA*+ *tcdB*+ *cdtA*, and *cdtB*+), V (RT078- *tcdA*+ *tcdB*+ *cdtA*, *cdtB*+), and VIII (RT017- *tcdA*- *tcdB*+ *cdtA*, *cdtB*- [30]. An increased disease severity associated with *tcdA*+ *tcdB* toxinotypes isolates has been reported, but the recurrence rate is similar to that involving *tcdA*+ *tcdB*+ toxinotypes [33,34,35].

Spore formation by *C. difficile* is crucial to the survival and dissemination of bacteria in the environment [36]. The various spore layers from the outside include the exosporium, coat, outer membrane, cortex, germ cell wall, inner membrane, and core [37]. The core of the spore is surrounded by an inner membrane, a peptidoglycan germ cell wall, and a large cortex layer composed of spore-specific modified crosslinked peptidoglycans that must be hydrolyzed in the spore germination and protect the bacteria against ethanol and heat. The cortex layer is surrounded by the outer membrane and the coat, which protects against enzymatic and chemical agents. Finally, there is a last layer known as the exosporium [38]. The spore of *C. difficile* is resistant to desiccation, numerous disinfectants, ultraviolet light, and antibiotics, allowing long-term survival against environmental insults and efficient transmission from host to host [39].

## 4. Biofilm

In addition to the ability to sporulate and produce toxins, other *C. difficile* virulence factors are associated with R-CDI, such as biofilm formation [40]. A biofilm is a community of bacteria organized and includes a single or multiple species growing attached or unattached [41] to a biotic or abiotic surface [42]. In a biofilm, cells are wrapped by an extracellular matrix composed of extracellular polymeric substances (EPSs), including polysaccharides, extracellular DNA (eDNA), proteins, glycoproteins, and glycolipids [43,44,45,46]. Bacteria may use biofilm as a protective barrier against multiple environmental stressors, such as antibiotics [40].

### 4.1. Biofilm Composition

*C. difficile* can form a multilayered biofilm in vitro with variable thicknesses depending on the duration and growth conditions [45]. Evidence regarding the chemical composition of *C. difficile* biofilms suggests that the proportions of the components may vary between strains and under different growth conditions. In a study that included 102 *C. difficile* isolates from different ribotypes grown in brain–heart infusion broth supplemented (BHIS) for 48 h, the main components reported were proteins [47] (Table 2). In contrast, when a biofilm of strain R20291 was grown in BHIS containing 0.1 M glucose for 48 h, the main component reported was eDNA [42]. In these studies, different models were used, and the results and findings cannot be compared. Several studies have shown that biofilm produced by *C. difficile* comprises proteins, eDNA, and polysaccharides [48,49,50].

### 4.2. Biofilm Formation

The biofilm formation process is often divided into three stages: initial attachment, maturation, and separation or dispersion [62]. The initial attachment stage of a biofilm, in which bacteria adhere to a surface, is mediated by different factors, including adhesins, flagella, pili, and fimbriae. This binding is initially reversible but becomes irreversible during maturation [45,62,63,64].

The maturation of biofilm is characterized by the production of an extracellular matrix [63], which forms the scaffolds of the biofilm structure and allows for interactions between the cells [45,65]. Moreover, it favors changes in the metabolism in response to the oxygen and nutrient gradient according to the location within the biofilm [65]. Stimuli such as changes in the microenvironment, antibiotic administration [45,63,64], QS, cyclic diguanosine monophosphate (cyclic-di-GMP) levels [63], and interactions with other bacterial species can favor biofilm sloughing and dispersal to colonize other sites [45,64].

An in vitro model of *C. difficile* biofilm on abiotic surfaces has shown that the number of spores inside the biofilm increases with time during biofilm maturation. Up to day 14 of growth, spores are considered the predominant form of *C. difficile* in the biofilm and are surrounded by extracellular EPSs [66,67].

The ability of *C. difficile* to produce biofilm in vivo was first described in 2017, using a germ-free model of CDI in an infected mouse with strains R20291 (RT 027), P30 (RT 014/020), and 630∆*erm* (RT 012) and the mutant 630∆*erm cwp84*::erm [68] (Table 2). In this study, the colonization levels by *C. difficile* in the jejunum and ileum were 100-fold lower than in the cecum and colon on day 7, and *C. difficile* bacteria were distributed heterogeneously over the intestinal tissue, without contact with epithelial cells. Bacterial cells were localized inside and outside the mucus layer, irrespective of the strains tested. Most bacterial cells of *C. difficile* were entrapped in 3D structures overlaying the mucus layer. For the R20291 strain, the PS-II was detected in large amounts in the 3D structure, suggesting that at least the R20291 strain is organized in the mouse model in glycan-rich biofilm architecture, which maintains bacteria outside the mucus layer [68].

*C. difficile* biofilms have been studied in various in vitro models, including microplates [51], black polycarbonate membranes [67], T-flasks [51,66], microfermenters [46] chemostats [40], and with different mutant strains (Table 2). According to these models, the *C. difficile* biofilm composition and structure depend on the incubation time [66], strain [66,69], and growth rate [42,67].

Recently, the transcriptomic profile in a biofilm model of RT001 and 027 associated with R-CDI and not associated with recurrent (NR)-CDI was analyzed to identify genes that may favor the recurrence using microarrays. In this study, *CAJ70148*, *CAJ68100*, *CAJ69725*, and *CAJ68151* genes were differentially expressed in biofilm in strains associated with R-CDI; thus, they may support the biofilm favoring the recurrence of CDI [70].

### 4.3. Systems and Structures Associated with Adhesion and Biofilm Formation

Biofilm formation is a multistep and complex process, and approximately 20% of the *C. difficile* genome is expressed differently between biofilm and planktonic cells. Many of these genes participate in multiple pathways, and their expression changes according to the *C. difficile* strain, biofilm model conditions, biofilm stage, and incubation time [46]. Most genes involved in biofilm formation have been determined by directed mutation, its effect on increasing or reducing biofilm production, and the capacity to produce the disease [46].

#### 4.3.1. Cell-Wall Proteins

*C. difficile* produces proteins that mediate bacterial adhesion to host cells [71], mucus layers [68], and other bacteria or surfaces and allow for attachment and colonization. Most of these proteins belong to the cell-wall protein family (Cwp), and some are involved in the formation of biofilms [48,72].

In the Cwp family, Cwp84 cleaves components of the extracellular matrix in eukaryotes, such as fibronectin, laminin, and vitronectin [48,73], and cleaves SlpA, which is necessary for achieving a paracrystalline arrangement that envelops the bacterial surface and anchors S-layer proteins [74,75]. Within the same family, Cwp66 exhibits 56% similarity to the N-acetylmuramoyl-L-alanine amidase of *Bacillus subtilis* [44,76].

In an in vitro model of *C. difficile* biofilm on a polycarbonate filter, in the first stage of biofilm formation, some planktonic cells of *C. difficile* undergo autolysis to produce eDNA and cellular debris [67,77]. Cwp19, a peptidoglycan hydrolase, probably triggers this process with its lytic transglycosylase activity [77] or through the differential expression of toxin-antitoxin systems, such as the MazE-MazFTA [78], CD2517.1-RCd8, CD2907.1-RCd9, and CD0956.2-RCd10 systems [53,79,80,81].

#### 4.3.2. Quorum Sensing

QS is a complex communication system that allows bacteria to communicate with other cells from the same bacterial species or other species and is fundamental to the persistence, growth, and dispersion of bacterial cells [63]. QS in bacteria involves the production of self-secreted extracellular signaling molecules such as acyl homoserine lactones, autoinducers (AI), oligopeptides, diffusible signal factors, and autoinducing peptides (AIP) [82,83,84,85]. QS regulates cell-population density and regulates diverse bacterial processes, including *C. difficile* toxin production [86,87], activation of flagella [15,22,86], sporulation [26,88], and biofilm formation [49,56]. The bacterial cell detects the increase in AIs and triggers a signaling cascade that alters the gene expression [56]. The QS pathways partially identified in *C. difficile* are LuxS and the accessory gene regulator (Agr) system [89].

##### Lux S

The *luxS* is a 53 bp gene that encodes LuxS, an AI-2 synthase [90]. LuxS participates in the recycling pathway of methionine [91] by cleaving S-ribosylhomocysteine to form homocysteine [90] and 4,5-dihydroxy-2,3-pentanedione (DPD), an unstable furanone that spontaneously cyclizes into several different forms [90], collectively known as AI-2, a group of potent cross-species QS-signaling molecules [56,90,91]. The effect of *luxS* on biofilm production has been demonstrated in the *luxS* mutant strain R20291, which reduces biofilm production in vitro [42], with no difference in the number of cells or the production of spores detected. Additionally, it has been demonstrated that the biofilm phenotype can be restored by adding 100 nM DPD [56].

When the transcriptomic profile of mutant *luxS* was compared against the wild type, 21 genes were found to be differentially expressed: 2 were upregulated (2 involved in trehalose and one in glucose metabolism), and 18 were downregulated (2 prophage regions: CDR20291_1415–1464 and CDR20291_1197–1226). Transcriptomic analyses also revealed the downregulation of prophage loci in the luxS mutant biofilms compared to the wild type. The detection of phages and eDNA within biofilms suggests that DNA release by phage-mediated cell lysis contributes to *C. difficile* biofilm formation [56].

##### Agr

AIs can be synthesized through the Agr system, which encodes a transcriptional response regulator (AgrA), a protease (AgrB), a sensor histidine kinase (AgrC), and a signaling pre-peptide (AgrD). In *C. difficile*, three Agr systems, Agr1, Agr2, and Agr3, have been identified, with a different type of organization of the genes. It has been reported that *C. difficile agr1* mutants affect the sporulation, motility, and toxin production of *C. difficile* strains 630 and R20291 [92].

In addition to *agr1*, *agr2* locus is present in *C. difficile* strains (R20291, RT001, and RT017 [92]. *C. difficile* Agr2 is similar to *S. aureus* with genes of the AGR system (*agrACDB* arranged) in the inverse order of those found in the *S. aureus* Agr system (*agrBDCA*) 85 [93]. Most strains of *C difficile* exhibit an incomplete *agr* locus that contains *agrDB* and is called the *agr1* locus. A third locus (*agr3)* has also been found in the *C. difficile* strains NAP07, NAP08, and QCD-23m63, all of which also encode *agr1* [94]. The *C. difficile* Agr3 system consists of *agrB3*, *agrD3*, and *agrC3* [95]. It has been reported that Agr3 seems to be encoded by a *C. difficile* bacteriophage phiCDHM1, suggesting the transmission between *C. difficile* strains [94]. The role of the *Agr* system in the biofilm formation of *C. difficile* is still poorly understood.

Gram-positives have a special QS system in which the receptor interacts with its cognate signaling peptide. The receptors are either Rap phosphatases or transcriptional regulators and integrate the protein family RNPP from Rap, Npr, PlcR, and PrgX. These systems control sporulation, virulence, biofilm formation, and the production of extracellular enzymes [96].

#### 4.3.3. Cyclic di-GMP

It has been reported that, when biofilm and planktonic cells are compared, 751 genes are differentially expressed, with 338 upregulated in biofilm. These genes are involved in metabolic pathways, including T4P production: *pilA1* [97] cell-wall biosynthesis [98]; and the production of a diguanylate cyclase, which is associated with the synthesis of the second messenger cyclic di-3′,5′-guanylate (cyclic-di-GMP) [46,51,53,59,99]. Cyclic di-GMP controls several cellular functions of *C. difficile*, including virulence, motility, and adhesion [59,100], and it participates in the posttranscriptional regulation of biofilm formation [51]. Cyclic di-GMP also upregulates 42 genes, with 37 involved in chemotaxis and flagellar motility [53]. Additionally, it favors a sessile lifestyle by modulating the attachment of cell-wall proteins to peptidoglycans [53,59,101]. Cyclic di-GMP represses the major operon *flgB* through a type I cyclic-di-GMP riboswitch (Cdi1-3), which reduces motility [51,53,59,101,102].

#### 4.3.4. Type IV pili

It has been reported that type IV pili (T4P) favor biofilm production by adhesion to abiotic or biotic surfaces, colonization [44,103], twitching motility [59], and microcolony formation [104]. Generally, regulation via cyclic-di-GMP acts as an “on switch” for T4P genes [59], which are differentially expressed during biofilm growth.

In a 7-day biofilm (*C. difficile* R20291 strain), it has been reported that *pilA1*, *pilJ*, and *pilW* are highly expressed [97], with *pilA1* being the most upregulated gene compared to planktonic cells. Furthermore, the *piA1* mutant formed a thinner biofilm with lower biomass [44,46].

#### 4.3.5. Flagella

The flagellum is a rotating semi-rigid helical filament anchored within the bacterial membranes and driven by the influx of protons or Na+ ions. It allows bacteria to move within fluid environments, including through liquid films on surfaces [105]. Flagella have been found to play a role in flagellum-dependent swimming motility, providing the organism an advantage over non-flagellated strains [106]. Flagella functions as an adhesin and a type III secretion system that can regulate virulence factors and is essential for the induction of proinflammatory responses and the invasion of host cells [106,107].

An in vitro model showed reduced biofilm production in the *C. difficile* R20219 strain (*fliC* mutant) after five days of incubation on a BHIS medium containing 0.1 M glucose 42. A second in vitro biofilm model in glass coverslips after 7 days in BHI broth showed that the expression of *fliC* significantly decreased in biofilm compared to planktonic cells [44]. It has been proposed that flagella-mediated motility is required in the late stages of biofilm formation [69].

## 5. The Role of Spores and Biofilms in CDI and R-CDI

It has been proposed that the internalization of spores in intestinal cells during infection and germination under favorable conditions could favor the recurrence or persistence of the infection [108]. Bacterial biofilm formation has been associated with chronic and recurrent infections [40,109]. The protection provided by the biofilm structure prevents the entry of antibiotics and favors the formation of persister cells [110]. Moreover, the exchange of resistance genes through plasmids, eDNA, and phages may be a source of recurrence cases or therapeutic failure [40,111].

It has been reported that *C. difficile* biofilm can survive treatment with vancomycin (VAN), thus promoting recurrence [40]. In a previous in vitro study, a biofilm formed with normal gut microbiota was treated with clindamycin and *C. difficile* spores. *C. difficile* was incorporated into the biofilm of the normal microbiota, and greater numbers of spores, vegetative cells, and toxins were detected. Next, *C. difficile* was depleted using VAN, but an increase in spores, vegetative cells, and toxins was observed 100 days later, which strongly suggested recurrence [40].

Furthermore, inside the biofilm of *C. difficile* strains associated with recurrence, increased expressions of the QS gene (*agrD1*), adhesion gene (*cwp84*), and sporulation pathway genes (*sigH*, *spo0A*) were found [112]. In another study, mice infected with the asporogenic phenotype of *C. difficile* (R20291Δ*spo0A*) treated with VAN did not present recurrent CDI. Furthermore, the R20291Δ*spo0A* demonstrated decreased biofilm formation in vitro [113].

## 6. Biofilm and Antibiotic Resistance

The clinical practice guideline by the Infectious Diseases Society of America and Society for Healthcare Epidemiology of America recommended, in 2021, the use of fidaxomicin (FDX), VAN, or metronidazole (MTZ) to treat CDI episodes. In the first or second recurrence, FDX or VAN is recommended, along with a pulsed regimen plus bezlotoxumab [4]. However, no treatment is 100% effective at reducing R-CDI [114].

Therapeutic failure in R-CDI has been associated with the inadequate concentration of drugs in the intestinal tract to eliminate *C. difficile* [114,115], the metabolism and inactivation by the microbiota [116], the ability to generate metabolically inactive spores, and the antibiotic resistance of a biofilm [112].

The capacity of antibiotics to kill planktonic cells differs from their ability to kill cells in a biofilm, and the effects of antibiotics on biofilms depend on the type and concentration of the antibiotic used [117,118]. The presence of eDNA, the development of persister cells, and the abundance of EPSs prevent the penetration of antibiotics into biofilms, limiting the activity of antibiotics on aging cells [117,118]. Moreover, the ability of an antibiotic to penetrate a biofilm depends on the bacterial species or strain, the antimicrobial agent, and the growth conditions of the biofilm [118].

The role of biofilm formation in resistance to antibiotics has been reported. In one of these studies, the susceptibility of *C. difficile* (planktonic cells) to VAN and MTZ was evaluated in 123 isolates. The production of biofilms was also assessed, and 44% were strong biofilm producers [119]. Most isolates with reduced susceptibility to MTZ were strong biofilm producers (63%, 17/27), whereas 22.2% were non-biofilm producers. Furthermore, 72.7% of isolates with reduced susceptibility to VAN were strong biofilm producers, and 51% of isolates susceptible to VAN were found to be non-biofilm producers [119].

It is important to evaluate the activity of the antibiotics used to treat CDI in *C. difficile* biofilms to reduce the recurrence rate and clinical duration of the primary infection. Table 3 summarizes the activity of the most-used antibiotics and a clinical phase III drug called surotomycin (STM) against *C. difficile* biofilms [2].

### 6.1. Fidaxomicin

FDX inhibits the RNAP [128] and is associated with a significantly lower recurrence rate of CDI [115]. An observational cohort study in 2013–2016 that included 271 patients with CDI revealed that FDX was better than MTZ, VAN, and MTZ plus VAN for the prevention of R-CDI. In this study, even using FDX, 6.3% of patients had R-CDI, and 33.3% had a second recurrence [129].

A previous study reported that FDX distributes into *C. difficile* biofilms in minutes [120]. Furthermore, subminimal inhibitory concentrations (sub-MICs) of FDX (0.03x, 0.25x, and 0.50x MIC) exhibit a dose-dependent lowering effect on biofilm formation. For example, at 0.50x MIC, FDX reduced planktonic growth and biofilm formation [121]. In this study, the biofilm structures were thick, with reduced biomass at sub-MICs of 0.09x and 0.25x MIC of FDX. FDX was more effective than MTZ at reducing *C. difficile* spore counts within biofilms, and this may explain its effectiveness with R-CDI [120]. In another study, 0.125x MIC of FDX was added during the stationary phase of a culture of *C. difficile* (*C. difficile* UK-14 and ATCC 43255 strains), and pre-existing spores were not eliminated at 0.25x but prevented the production of spores [122]. Finally, sub-MICs of FDX have been found to induce a dose-dependent reduction in the number of viable spores, with 0.25x MIC reducing *spo0A* transcription in 9689 and 5325 strains [123].

The significant advantage of FDX is the reduction of the risk of R-CDI when compared to VAN [114,115,130,131], probably due to declining biofilm production, planktonic cell-killing activity, and reduction in spore count.

### 6.2. Vancomycin

VAN is a tricyclic glycopeptide that binds to the D-Ala-D-Ala moiety of monomers. After this binding process, the monomers cross the cell membrane, interrupting the process of cell-wall synthesis [132]. The biofilm minimum inhibitory concentrations (BMICs) of *C. difficile* were analyzed in 102 clinical isolates, and we detected reduced susceptibility to VAN (91.0%) and linezolid (89.21%). It was also found that the BMIC was 100 times higher than the MIC for VAN and 20 times higher for linezolid in the biofilm state than in planktonic MICs [112].

The production of biofilms and the susceptibility to VAN and MTZ in 123 *C. difficile* isolates have been assessed: 53.6% were biofilm producers, of which 44% were strong biofilm producers [119]. Most isolates with reduced susceptibility to MTZ were strong biofilm producers (63%, 17/27), whereas 22.2% were non-producers. Furthermore, 72.7% of isolates with reduced susceptibility to VAN were strong biofilm producers, and 51% of isolates susceptible to VAN were found to be non-producers [119]. Many studies have demonstrated that the sub-MICs of VAN do not inhibit biofilm formation or planktonic growth. However, only 0.25x MIC of VAN has been found to significantly reduce the biomass of biofilms without the expression of genes involved in biofilm production, including *pilA1*, *cwp84*, *luxS*, *dccA*, and *spo0A* [121].

Furthermore, the effects of DNase and proteinase K on vegetative cells and spores in the biofilm and other states have been investigated [51]. A reduced viable vegetative cell count in intact biofilms was observed with 12.5 mg/mL VAN. In contrast, disruption with DNase combined with VAN treatment reduced the vegetative cell count to 0.68% of the untreated control. Furthermore, treatment with proteinase K combined with VAN reduced the viable vegetative cell count to only 72.8% of that of the untreated biofilm. Therefore, combining DNase and VAN is more efficient than using either alone to reduce the viable vegetative cell count in a biofilm [51]. Furthermore, VAN does not affect spore viability, irrespective of biofilm disruption [51].

### 6.3. Metronidazole

The mechanism of action of MTZ has not yet been fully elucidated. MTZ crosses the target cell membrane via passive diffusion [133]. In this process, the nitro group of the molecule is reduced to nitro radicals by ferredoxin or flavodoxin, generating toxic metabolites. These metabolites, such as N-(2-hydroxyethyl)-oxamic acid and acetamide, can react with DNA and form adducts with guanosine [134].

MTZ increases in vitro biofilm formation at 0.25x and 0.5x MICs in the *C. difficile* ribotype 010 and produces a thick biofilm composed of layered aggregates [121,126].

The effect of MTZ on the sporulation of *C. difficile* in liquid broths has been analyzed (ribotypes 001, 012, 037, 027, 018, 027, 014, and 018). However, the results revealed that, at 0.5x MIC, the level of sporulation was not affected in most strains [124]. In a similar study, sporulation was not inhibited in NAP1/BI/RT027 and CD196 strains growing with 0.25x MIC of MTZ [122]. In contrast, 0.25x, 0.125x, and 0.0625x MICs of MNZ were found to stimulate sporulation in the epidemic 5325 strain at 48 h (~2 log increase). However, all subinhibitory concentrations of MTZ suppressed spore formation in strain 9689 [123].

A proportion of MTZ administered orally may be degraded by the microbiota, yielding sub-MIC doses. It has also been found that the germination of spores and the production of cytotoxins increase when the MIC decreases in a chemostat model with strains RT001 and 027 [116]. Similarly, cells of the toxigenic *C. difficile* strains BI17 (PCR ribotype 027) and J9 grown as a biofilm have exhibited tolerance to MTZ concentrations as high as 100 μg mL^−1^. However, antibiotic concentrations above 1 μg mL^−1^ inhibited the same strains in liquid cultures, suggesting a 100-fold increase in resistance to this drug in the form of a biofilm [67].

## 7. Conclusions

The evidence suggests that biofilm formation in *C. difficile* is affected by multiple factors, including environmental and bacterial virulence factors, and that biofilm formation affects the recurrence of CDI.

Biofilm plays several roles that may favor recurrence, for example, allowing spores to remain inside the biofilm matrix and protecting the vegetative cells from the activity of antibiotics, even at concentrations higher than therapeutic doses.

Most antibiotics recommended for the treatment of CDI do not have activity on spores and do not eliminate biofilm. This makes it challenging to eradicate *C. difficile* in the intestine, thus complicating antibacterial therapies and allowing non-eliminated spores to remain in the biofilm, increasing the risk of recurrence.

According to one perspective, the deep analysis of molecules participating in QS may provide potential QS inhibitors that may be therapeutic alternatives.

Finally, no direct evidence links biofilm to CDI recurrence, so further studies are needed.

## Figures and Tables

**Figure 1 microorganisms-11-02525-f001:**
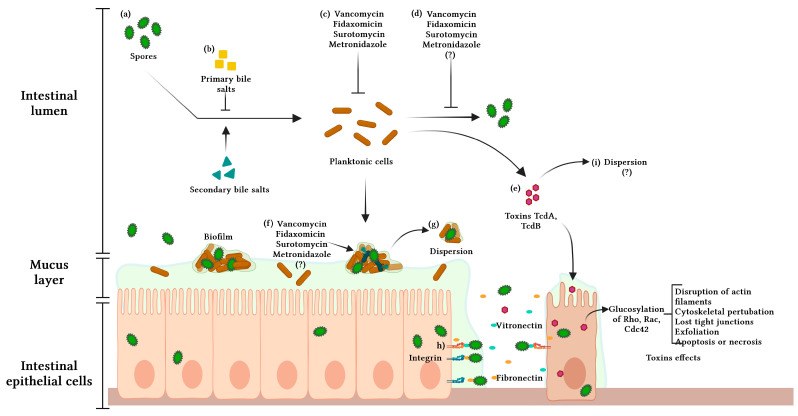
Pathophysiology of *Clostridioides difficile* infection. (**a**) The infection begins with the consumption of spores. (**b**) Spores in the intestine can interact with primary bile salts that prevent its germination or with secondary bile salts and germinate. (**c**) Antibiotics kill the planktonic cells of *C. difficile*. (**d**) Antibiotics can inhibit the formation of spores. (**e**) Planktonic cells secrete toxins that affect the cytoskeleton of the intestinal cell and adhere to the colon mucous layer. (**f**) Planktonic cells can form a biofilm-like structure, hindering the action of antibiotics on the cells embedded in the biofilm. (**g**) Biofilm can be dispersed to other sites and colonized. (**h**) Spores can be internalized in intestinal epithelial cells. (**i**) It has been suggested that spores can disperse to other sites to colonize. Adapted from “Structure of Mucosal Barrier” from BioRender.com (2022). Retrieved from https://app.biorender.com/biorender-templates (accessed on 9 May 2023).

**Table 2 microorganisms-11-02525-t002:** Effects of gene mutations in *C. difficile* biofilm.

Strain	Biofilm Model	Biofilm Effect	Ref.
**R20291::*fliC*430s**	24-WPP; BHISB + 0.1 M glucose; 120 h	Decreased biofilm production on day 5	[42]
**R20291::*luxS*161a**	24-WPP; BHISB + 0.1 M glucose; 72 h	Decreased biofilm production. Unable to form even a bacterial monolayer	[42]
**R20291::*sleC*128a**	24-WPP; BHISB + 0.1 M glucose; 72 h	Form thick biofilm-like structures	[42]
**R20291::*spo0A*178a**	24-WPP; BHISB + 0.1 M glucose; 24, 72, 120 h	Decreased biofilm formation. Form uneven and thick biofilm-like structures. Cellular form filamentous structures	[42]
**R20291Δ*cwp84***	24-WPP; BHISB + 0.1 M glucose; 24 h	Decreased in biofilm production	[42]
**R20291- *cdtA* and *cdtB*, *cwlD and cwlD* ***	24-WPP; BHISB + 0.1 M glucose; 24, 72, 120 h	Unchanged	[42]
**R20291Δ*pilA1***	Glass jars with 8 mm glass beads and glass coverslips; BHISB; 24 h	Reduced thickness and biomass. Decreased live cell count and a decreased tendency to aggregate	[44]
**630Δ*erm*::CD2214, Erm^R^ Tm^S^**	24-WMP; TYt, 48 h	Denser with several short-rod bacteria and smaller micro-aggregates	[46]
**630Δ*erm*::*pilA1*, Erm^R^ Tm^S,^ 630Δ*erm*::CD2831**	24-WMP; TYt, 24 h	Unchanged	[46]
**630Δ*erm* Δ*pilA1* (pRPF185 P_tet_ *dccA*), Erm^R^ Tm^R^**	96-well plate; TYt + anhydrotetracycline; 24 h	Form a dense and homogeneous, carpet-like biofilm, slightly decreased biomass	[46]
**630Δ*erm*ΔCD2831 (pRPF185 P_tet_ *dccA*), Erm^R^ Tm^R^**	96-well plate; TYt + anhydrotetracycline; 24 h	Form a dense and homogeneous, carpet-like biofilm without change in biofilm production	[46]
**630Δ*erm*Δ*cwp84* and R20291Δ*cwp84***	24-WPP; BHISB + 1.8% glucose; 72 h	Increased 72-fold. Denser and thicker. Protein abundance in biofilm was altered	[48]
**630Δ*erm*::2831, 630Δ*erm*::0183, 630Δ*erm*::3392, 630Δ*erm*::*CbpA***	24-WMP; BHISB; 24 h	Decreased biofilm production	[51]
**630∆*PEPP-1***	24-WMP; BHISB, 24 h	Unchanged	[51]
**630∆*PEPP-1*(P*_tet_2831*), 630∆*PEPP-1*(P*_tet_3246*)**	24-WMP; BHISB, 24 h	Increased biofilm production	[51]
**R20291::*spo0A***	24-WMP or 24-WMP with coverslips; BHISB, 72 and 144 h	Decrease in depth and breadth of the biofilm. Decrease the number of spores	[51]
**JIR8094:: *lcpB***	24-WPP; BHISB + 1.8% glucose; 72 h	Robust biofilm	[52]
**JIR8094:: *lcpA***	24-WPP; BHISB + 1.8% glucose; 72 h	Unchanged	[52]
**630Δ*erm* (pRPF185 P_tet_ *dccA*), Tm^R^**	96 well polystyrene plate; TYt, 48 h	Increased biofilm production. Biovolume increase of 1.6-fold. Highly homogeneous and dense	[46,53]
**R20291::*lrp,* 630Δ*erm*::*lrp***	24-WPP; BHISB + 0.1 M glucose; 72 h	Unchanged	[54]
**R20291::*lexA*238a**	24-WPP; BHISB + 20 μg/mL lincomycin: 24 h	Increased biofilm production	[55]
**20291::*luxS*161a**	24-WPP; BHISB + 0.1 M glucose; 24 y 72 h	Decreased biofilm production	[56]
**630Δ*erm*::*dnaK*723a**	96-well flat-bottom polystyrene plate; BHISB + 0.9% glucose; 24, 48 and 72 h	Increased biofilm production	[57]
**630Δ*erm*::CD1687, JIR8094::*codY,* JIR8094::*ccpA,* 630Δ*erm*::*spo0A***	24-WPTCTP; BHISB + 100 mM glucose + 240 µM DOC; 48 h	Decreased biofilm production	[58]
**630Δ*erm*::CD1688, 630Δ*erm*::*sigB,* 630Δ*erm*::*sigE,* 630Δ*erm*::*sigF***	24-WPTCTP; BHISB + 100 mM glucose + 240 µM DOC; 48 h	Unchanged	[58]
**630Δ*erm*::*cwp19***	24-WPTCTP; BHISB + 100 mM glucose + 240 µM DOC; 48 h	Failed to form a biofilm	[58]
**R20291Δ*cmrR***	24-WPP; BHISB + 1% glucose + 50 mM sodium phosphate buffer; 24 h		[59]
**R20291Δ*cmrT***	24-WPP; BHISB + 1% glucose + 50 mM sodium phosphate buffer; 24 h	Unchanged	[59]
**630Δ*erm*Δ*prkC***	24-WMP; BHISB + 0.1 M glucose + polymyxin B (20 μg·mL^−1^) or DOC (0.01%); 48 h	Produce 6- and 10-fold more biofilm than the WT	[60]
**630∆*erm*∆*pilW,* 630∆*erm*Δ*pilA1,* 630∆*erm*ΔT4P*2* cluster, 630∆*erm*Δ*sinR,* 630∆*erm*ΔCD630_08650, 630∆*erm*Δ*cwp29,* 630∆*erm*Δ*cysk,* 630∆*erm*Δ*agrBD,* 630∆*erm*ΔluxS, 630∆*erm*Δ*hprk,* 630∆*erm*Δ*fumAB*, 630∆*erm*Δ*nanEAT*, 630∆*erm*Δ*prdB*, 630∆*erm*Δ*fur*, 630∆*erm*Δ*rex***	24-WPTCTP; BHISB + 100 mM glucose + 240 µM DOC; 48 h	Unchanged	[61]
**630∆*erm*Δ*bcsA,* 630∆*erm*Δ*fliC***	24-WPTCTP; BHISB + 100 mM glucos e+ 240 µM DOC; 24 or 48 h	Unchanged	[61]
**630∆*erm*Δ*sigD***	24-WPTCTP; BHISB + 100 mM glucose + 240 µM DOC; 24 h	Unchanged	[61]
**630∆*erm*ΔT4P1 cluster, 630∆*erm*Δ*cdsB* 630∆*erm*Δ*spo0A,* 630∆*erm*Δ*sigH,* 630∆*erm*Δ*sigL,***	24-WPTCTP; BHISB + 100 mM glucose + 240 µM DOC; 48 h	Decreased biofilm production	[61]
**630∆*erm*Δ*ptsI***	24-WPTCTP; BHISB + 100 mM glucose + 240 µM DOC; 48 h	Induction of biofilm abolished by DOC	[61]

DOC, deoxycholate; BHISB, brain–heart infusion supplemented with 5 g/L yeast extract and cysteine 0.1% broth; TYt broth, tryptone yeast extract broth supplemented with 0.1% sodium thioglycolate; 24-WPP, 24-well polystyrene plate; 24-WMP, 24-well microtiter plate; 24-WPTCTP, 24-well polystyrene tissue culture-treated plate; *, not specified.

**Table 3 microorganisms-11-02525-t003:** Effects of antibiotics on *C. difficile* biofilms, spores, and planktonic cells.

Effect on Biofilm	Effect on Spores/Planktonic Cells	Changes in Gene Expression
**Fidaxomicin**		
Penetrates biofilms within 2 min [120]. 0.03x–0.25x MICs exhibit a dose-dependent inhibitory effect on biofilm formation [121]. 0.09x and 0.25x MICs cause thickness and biomass reduction [121]. 0.50x MIC decreases vegetative cell growth and biofilm formation [121]. 25x MIC decreases the spore count and kills vegetative cells within mature biofilms [120].	*Spores* 0.25x and 0.125x MICs during the stationary phase prevent the production of spores. 2x MIC decreases the outgrowth of vegetative cells [122]. *Planktonic cells* 0.25x MIC reduces viability [123].	*fliC* expression increases, but no change occurs in the expression of *pilA1*, *cwp84*, *luxS*, *dccA*, and *spo0A* [121]. 0.25x MIC decreases *spo0A* transcription [123]. No accumulation of *spoIIR* or *spoIIID* mRNA occurs [122]. In the non-biofilm state, it suppresses the expression of both *tcdA* and *tcdB*, with maximal repression at 1/4x MIC [123].
**Vancomycin**		
0.25x MIC is associated with reduced biomass [121]. 12.5 mg/mL reduces the viable vegetative cell count in intact biofilms with an enhanced effect by adding DNase [51]. It does not affect spore viability, irrespective of biofilm disruption [51].	*Spores* 0.5x MIC may affect the spore count [124]. 0.25x and 0.125x MICs do not affect sporulation during the stationary phase [122]. 0.25x and 0.125x MICs reduce spore production in 48 h cultures [123]. 2.5x MIC inhibits the outgrowth of vegetative cells and does not affect spore germination [125]. *Planktonic cells* It only inhibits the growth of vegetative cells [122].	0.25x MIC does not change the mARN expression of *pilA1*, *cwp84*, *luxS*, *dccA*, and *spo0A* [121]. In a biofilm state, 0.5x MIC increases the transcription of *tcdA* and *tcdB* toxins [124].
**Metronidazole**		
0.25x and 0.5x MICs increase in vitro biofilm formation. It stimulates the production of a thick biofilm composed of layered aggregates and influences extracellular matrix production ^a^ [121,126].	*Spores* 0.5x MIC does not affect sporulation [124]. 0.25x MIC does not inhibit sporulation [122]. 0.25x–0.125x and 0.0625x MICs stimulate sporulation in strain 5325 [123]. 0.25x–0.125x and 0.0625x MICs suppress spore formation in strain 9689 [123].	
**Surotomycin**	*Planktonic cells*	
It penetrates *C. difficile* biofilms in less than one hour and starts accumulating. It exhibits a disruptive activity on biofilm structure at 24 h [120]. 100x, 50x, and 25x MICs kill vegetative *C. difficile* strain ATCC BAA-1382 within biofilms in vitro [120].	8x and 80x MICs kill vegetative exponential-phase cells. 80x MIC kills stationary-phase cells [127].	

MIC, minimum inhibitory concentration; VAN, vancomycin; MTZ, metronidazole; FDX, fidaxomicin. ^a^ Contradictory evidence.

## Data Availability

Not applicable.

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
