# Peer review of "Review of the Impact of Biofilm Formation on Recurrent Clostridioides difficile Infection"

_microorganisms, 2023, doi:10.3390/microorganisms11102525_

Round 1

Reviewer 1 Report

The manuscript entitled "Impact of biofilm formation on recurrent Clostridioides difficile infection" has been kindly reviewed. Authors examined the activity of an antibiotic on biofilm, its association with recurrence, and the challenges of treating C. difficile infection when the bacteria form a biofilm. It is a much interesting review, however, some revisions should be carefully treated before further evaluation. My main concerns have listed as follows:

(1) Title: As it is a review, suggestion would throw to typical words, such as a review, a systematic review.

(2) Abstract: Rewrite or enrich this section, as its current expression did not tell me the background and sigificance of the current review.

(3) Introduction: It is simple here to introduce the work, so, add some recent publications to enrich them.

(4) 4.3.2 Quorum sensing, there many kinds of QS which may occur in CDI, please add others, like AI-1, to enrich them.

(5) Line 416: As a review, some perspectives and prospects should be provided, not just conclusion.

(6) Pay more attention to the form of reference, as the current is not uniform to the journal requires.

Author Response

Dear reviewer, 

Please see here the responses to your comments. We appreciate your valuable review.

Title: As it is a review, the suggestion would throw to typical words, such as a review a systematic review.

Response:  The title of the manuscript was changed to Review of the impact of biofilm formation on recurrent Clostridioides difficile infection

Abstract: Rewrite or enrich this section, as its current expression did not tell me the background and significance of the current review.

Response: The abstract section was rewritten

Introduction: It is simple here to introduce the work, so, add some recent publications to enrich them.

Response: The introduction was enriched by including some recent publications.

Quorum sensing, there many kinds of QS which may occur in CDI, please add others, like AI-1, to enrich them.

Response: We gave more details to quorum sensing and enriched the section.

As a review, some perspectives and prospects should be provided, not just a conclusion.

Response: As a perspective, the deep analysis of molecules participating in QS may provide potential QS inhibitors rather than therapeutic alternatives. This sentence was added to the new manuscript.

Pay more attention to the form of reference, as the current is not uniform to the journal requirements.

Response:  The references were carefully reviewed

regards

Reviewer 2 Report

The authors provided a comprehensive review on the Impact of biofilm formation on recurrent Clostridioides difficile infections. 

It is well organised and very thorough, including Pathophysiology of C. difficile infection, Virulence factors: Toxins and spores, the activity of an antibiotic on biofilm and the challenges of treating a CDI when the bacteria form a biofilm, molecular mechanisms of biofilm formation and its association with CDI recurrence. 

A few minor typos found:

Abstract:

Line 11: "C. difficile infection". Suggest spelling out "C. difficile" as  "Clostridioides difficile" when first mentioned. 

Line 18: R-CDI. Please spell out as Recurrent CDI (R-CDI)

Conclusion:

Line 427. “Finally, no direct evidence linking biofilm to CDI recurrence is needed, so further studies are needed.” The first part “is needed" is not make sense, might be a typo. Suggest change it to "Finally, no direct evidence links biofilm to CDI recurrence, so further studies are needed."

Author Response

Dear revisor, Please see here the response to your comments

Line 11: "C. difficile infection". Suggest spelling out "C. difficile" as  "Clostridioides difficile" when first mentioned. 

Response: The sentence was corrected to: Clostridioides difficile infection (CDI) may recur in approximately 10–30% of patients.

Line 18: R-CDI. Please spell out as Recurrent CDI (R-CDI)

Response: The sentence was corrected to: For the initial treatment of CDI episodes, antibiotics (alone or combined) —including fidaxomicin, vancomycin, or metronidazole—are recommended, and in Recurrent-CDI, fidaxomicin or vancomycin are advised.

Conclusion:

Line 427. “Finally, no direct evidence linking biofilm to CDI recurrence is needed, so further studies are needed.” The first part, “is needed" is not make sense, might be a typo. Suggest change it to "Finally, no direct evidence links biofilm to CDI recurrence, so further studies are needed."

Response: The sentence was changed as suggested by the reviewer.

Regards